# Thymus Degeneration in Women and the Influence of Female Sexual Hormones on Thymic Epithelial Cells

**DOI:** 10.3390/ijms26073014

**Published:** 2025-03-26

**Authors:** Meiru Zhou, Yaoying Shu, Jianli Gao

**Affiliations:** School of Pharmaceutical Sciences, Zhejiang Chinese Medical University, Hangzhou 310053, China; 202321124011185@zcmu.edu.cn (M.Z.); 202211124011146@zcmu.edu.cn (Y.S.)

**Keywords:** thymus degeneration, female, regulation mechanism, hormone

## Abstract

The thymus is a central immune organ for T cell development and plays an extremely important role in immune and aging. The unique physiological processes that occur in women, such as the menstrual cycle, pregnancy, and menopause, contribute to sexual dimorphism in thymic immunity. Thymic epithelial cells (TECs) are key stromal cells that affect thymus development and degeneration. Interestingly, TECs in women have stronger proliferation potentiality and ability for output of T cells than those in men. In comparison to men, women exhibit higher susceptibility to autoimmune disease, which can be attributed to lower AIRE expression in the female thymus, which is influenced by fluctuating hormone levels. In this review, we summarize the principles of female thymus regulation by hormones, particularly the influence of female sex hormones in the development and function of TECs, as well as the underlying mechanisms, with the aim of providing new ideas and strategies to inhibit or slow down female thymus degeneration.

## 1. Introduction

The thymus is an important immune organ in the body that develops from the endoderm of the third and fourth pharyngeal capsules during embryonic development [1]. As a major, central immune organ in the body, it is important for T cell differentiation, development, and maturation and plays an extremely important role in the body’s immune system. The thymus reaches its peak weight during puberty and subsequently undergoes atrophy. It is also the earliest organ to exhibit senescence. Following sexual maturation, the thymus undergoes degeneration and atrophy, resulting in a reduction in size, weight, and cell count, as well as decreased cell proliferation and increased apoptosis [2]. Previously, it was thought that aging caused problems in the thymus of middle-aged and elderly people, but new evidence shows that these changes are quantitative, not qualitative, and that people up to 107 years of age can still produce new T cells and have thymic tissue [3].

The sexual dimorphism of thymus development and degeneration is obvious. Overall, throughout the entire life cycle of humans, the volume of the thymus in men is greater than that in women of the same age (<49 years, approximately 80–85% of males), and there is relatively faster degeneration in men [4,5]. Especially during pregnancy and menopause, women’s thymuses undergo special changes, specifically rapid degeneration during pregnancy and childbirth, rapid regeneration after childbirth, and thymus hypertrophy during menopause. Some pathological changes ultimately result in a decline in the function of T lymphocytes incubated by the thymus, restriction of memory T lymphocyte replication, obstruction of peripheral T lymphocyte recycling to the thymus pathway, and a significant reduction in T cell output capacity.

Histologically, the thymus can be divided into two different regions, the cortex and medulla, containing different stromal cells and thymocytes in different stages of maturation [6]. The thymic microenvironment is mainly composed of thymic stromal cells (TSCs), extracellular matrix (ECM), and hormones and cytokines secreted by the stromal cells, providing the most suitable environment for the development of thymocytes [7,8]. The stromal cells of the thymus are composed of thymic epithelial cells (TECs), fibroblasts, endothelial cells, macrophages, dendritic cells, neural crest–derived pericytes, and other mesenchymal cells [9,10]. Among these, TECs provide a unique microenvironment for multiple stages of T cell development [11].

Studies have demonstrated that cTECs exhibit better proliferation potential in women compared to men [12], while factors such as menstruation, pregnancy, and menopause, influenced by women’s unique physiological features, affect thymus degeneration. Additionally, women are more prone than men to immune-mediated diseases like autoimmune and inflammatory disorders. Therefore, it is crucial to explore the regulation of and molecular mechanisms underlying female thymus development and atrophy to enhance overall immunity and delay immune system aging.

## 2. Thymus Degeneration in Women

### 2.1. The Overall Process of Thymus Degeneration in Women

The thymus displays sexual dimorphism in its age-related changes. According to clinical observations and research studies, hormones play a crucial role in thymic development during all stages, with women experiencing less pronounced thymus degeneration compared to men due to regulation by hormones (Figure 1A) [13,14]. (For ethical reasons related to human research, the most reliable data on the weight of the human thymus before the age of 30 and after the age of 70 are unavailable. Consequently, the relevant sections in Figure 1A are indicated by dashed lines for reference purposes only). Research conducted by J. G. Simpson et al. has revealed specific alterations in women compared to men, with cortical degeneration progressing in a manner that is particularly distinctive in women (Figure 1D) [15]. Meanwhile, studies have found that the average cross-sectional area of the female thymus increases from 696.90 mm^2^ at ages 40 to 49 to 706.86 mm^2^ at ages 50 to 59, with an extended cycle of overall atrophy lasting 10–20 years longer than that observed in men (Figure 1C) [4].

Estrogen is involved in the early development of the thymus. The embryonic period plays a crucial role in thymus development. Importantly, estrogen receptor (ER) expression during later stages of embryonic development closely correlates with formation of thymic tissue. The high expression level of ER at day 16 suggests its association with T cell differentiation [16]. The first stage of thymic degeneration in mice begins at 4–7 weeks of age, when the thymic cell structure is dramatically reduced [17]. Hormones may be involved in early-stage thymus development through the aromatase pathway [18].

A significant correlation exists between thymus size and neonatal body weight and length. Clinical data indicate that male neonates exhibit a higher thymic index compared to female neonates [19]. While the thymic weight of mice decreases from day 18 to 26 after birth, then steadily increases after day 26 in both sexes, with a longer increase period in females (day 46) than in males (day 35) [20]. In addition, there are sex-specific differences in vascular endothelial growth factor A (VEGFA) signaling within ECM fibroblasts and other mesenchymal cells. Male fibroblasts express more VEGFA and fibroblast growth factor (FGF) 7. CellChat predicts that male proliferating fibroblasts are enriched in FGF10, and only male vascular smooth muscle cells (VSMCs) express FGF18. These sex biases in fibroblast growth factor gene expression may contribute to the larger size of early postnatal male thymuses [21].

Following puberty [13], the increased secretion of estrogen exacerbates atrophy of the thymus, leading to a reduction in T cell subsets, with a disproportionate loss of double-positive (DP) cells and differentiation of thymic epithelial progenitor cells (TEPCs) accompanied by reduced turnover of TECs [13]. Additionally, female thymic lobes contain significantly higher numbers of DPs and cTECs [21]. In 8-week-old mice, female mice have slower medulla atrophy compared to male mice, with more DP cells but smaller double-negative (DN) and single-positive (SP) cells than male mice [22]. Elevated levels of estrogen directly inhibit the production of thymic precursors such as Flt3^+^ LSK (spectral Sca-1^+^ c-Kit^+^) and early thymic progenitors (ETPs), as well as the DN subpopulation within the bone marrow [23]. These findings suggest that high levels of female sex steroids affect both the thymus and the proliferation of hematopoietic cells in the bone marrow, including potential thymic progenitors.

Women exhibit a slower rate of thymus involution between the ages of 20 and 30 years than men [14]. This sexual dimorphism in thymus attenuation is likely attributable to variations in the hormonal environment. The low expression of Apolipoprotein D (APOD), a gene associated with hormone signaling, in female fibroblast populations suggests an earlier onset of thymic involution in males relative to females [21]. This period is the age of pregnancy, and pregnancy triggers a significant but temporary involution in the thymus that relies on TEC activity and plays a crucial role in enhancing reproductive fitness (Figure 1B) [12,24]. Additionally, pregnancy induces transient TEC-dependent involution of the thymus, enhancing reproductive fitness [12]. At 9 months old, female mice had significantly more cells in their thymus, especially DP cells, which show higher CD3 expression, indicating more successful *αβ* T cell receptor (TCR) pairings (Figure 1E) [25]. Female mice exhibit more efficient maintenance of TEC differentiation compared to male mice. There was an exclusive and concurrent increase in autoimmune regulator (Aire)^+^ cTEC/thymocyte ratio of middle-aged female mice with a decrease in Aire^+^ the medullary thymic epithelial cell (mTEC)/thymocyte ratio [26]. According to research findings, the Treg population expands in the late follicular phase of the menstrual cycle, while significantly decreasing in the following luteal phase [27]. The values tend to be consistently higher in females, suggesting that thymic output is prolonged at higher levels in women compared to men, lasting until the age of 50 [28].

The decrease in female sex organ function post-menopause results in a subsequent decline in hormone secretion levels, thereby diminishing the significance of hormone-induced degeneration of thymus as a contributing factor to female thymic aging. Araki et al. reported that CT scans have revealed differences in the appearance of the thymus in middle-aged and elderly men and women, which indicated that the average cross-sectional area of the thymus (anteroposterior diameter × transverse diameter) in women aged 50–59 increased to 706.86 mm^2^, compared to 696.90 mm^2^ in those aged 40–49. Their research suggests that women’s thymus atrophy process lasts 10 to 20 years longer than men’s [4]. Ovariectomy (Ovx) is a widely employed model for studying menopause. Research has revealed that Ovx induces a significant increase in thymus weight [29]. It is reasonable to speculate whether there exists a transient period of thymic hypertrophy during menopause, characterized by an extreme decline in estrogen levels and alterations in other hormones.

Additionally, a study found that older women have less pronounced steatosis but experience marked organizational changes in lymphoid tissue, including loss of the cortical–medullary demarcation line and disorganization of the perivascular space [30]. The study also observed a decrease in thymocytes and TEC activity.

### 2.2. Transient Physiological Thymus Degeneration During Pregnancy

During pregnancy, the thymus shrinks, with reduced volume and cortical size, and temporary medullary expansion occurs during severe hormonal changes. Thymocyte proliferation and output decrease, and T cell subsets are depleted [31,32,33]. Laan et al. found that pregnancy reduces major thymic lymphocyte populations, such as early T lymphocyte progenitor (TLP) and regulatory T cells (Tregs) [34]. The percentage of DN1 cells initially rises but later falls. This indicates a persistent decrease in TLP homing and a developmental blockage of DN1^−^DN2^−^ double-positive thymocytes [34]. Treg cells, which are beneficial for pregnancy, decrease in number as gestation progresses, especially in late stages [34]. Estriol at first-trimester concentrations reduces the number of certain myeloid dendritic cells, and kisspeptin at concentrations of the second to third trimester regulates the differentiation of thymic myeloid dendritic cells (DCs) and decreases some cell numbers [35]. Gestational acute thymic involution (ATI) is linked to high levels of progesterone-dominant hormones, which induce ATI through three mechanisms: inhibiting chemokine expression in thymic stromal cells, hindering TLP recruitment and migration; interacting with progesterone receptors (PR) in cTECs; and suppressing the secretion of soluble factors like interleukin (IL)-7 by thymic stromal cells. This inhibition leads to the suppression of DN cell proliferation and blocks their differentiation into DP cells, consequently hindering positive selection while promoting negative selection [24,34,36,37].

Postnatal thymus regeneration is primarily driven by cTECs, with forkhead box N1 (FoxN1) and its target genes like chemokine (C-C motif) ligand (*CCL*) *25*, chemokine (C-X-C motif) ligand (*CXCL*) *12*, *δ*-like ligand 4 (*DLL4*), cathepsin L (*CTSL*), and serine protease (*PRSS*) *16* playing a role in postpartum thymus regeneration. FoxN1 expression in cTECs and mTECs rises quickly postnatally, and its target genes are upregulated at 2 days, boosting thymocyte numbers and output [36]. During early regeneration, genes associated with T cell activation, proliferation, and differentiation are increased in cTECs, as are genes associated with MHC-I antigen presentation and protein hydrolysis, such as proteasome subunit beta (*PSMB*) *9*, *PSMB10*, and *PSMB11*. The proportion of MHC-II^high^ cTECs also increases, with a number of thymopoietic related genes being co-expressed and aiding in thymic regeneration [36]. In the late stage of thymus regeneration, lymphoid tissue inducer (LTi) and CD4^+^ cells enhance nuclear factor-κB (RANK) ligand (RANKL) expression, and exogenous RANKL treatment promotes thymus repair by increasing Ki67 expression in cTECs and mTECs [38].

### 2.3. Special Modifications in the Thymus of Postmenopausal Women

Menopause is marked by hormonal changes and aging. Post-menopause, reduced ovarian function leads to lower hormone levels, lessening the impact of hormone-related thymus aging. Key hormonal shifts include decreased estrogen, elevated follicle-stimulating hormone (FSH), and changes in progesterone, luteinizing hormone (LH), and testosterone (T) [39]. Menopausal women also show higher levels of proinflammatory cytokines like IL-1*β*, IL-8, and tumor necrosis factor (TNF) -*α* and lower levels of IL-20 [40].

The first discovery of sex differences in thymic involution was reported in 1975. Specifically, medullary atrophy follows a consistent linear pattern, whereas cortical degeneration exhibits a biphasic pattern characterized by phases of hypertrophy and accelerated senile degeneration around the age of 42 in females [15]. A study of CT scores found that menopausal women (40–49 years old) had higher thymus scores than younger women (30–39 years old), but there was a more significant decline post-menopause. CT scans showed that the mean thymus cross-sectional area in women aged 50–59 increased by 9.96 mm^2^ compared with the 40–49 age group [4]. Therefore, there may be transient regeneration of the thymus during menopause, wherein hormones play a crucial role. However, this phenomenon has likely been overlooked owing to human ethical constraints and the absence of long-term comparative studies of the thymus in both sexes.

Although ovariectomy (Ovx) and menopause result in minor differences in hormone levels, the former remains a widely adopted model for investigating menopause due to their numerous similarities. Following Ovx, there is a significant increase in thymus weight and cortical area [26,29,41]. It was observed that the levels of CD3^+^ and CD8^+^ in peripheral blood decrease in menopausal women, while the CD4^+^/CD8^+^ ratio increases. Similar findings have been reported in ovariectomized animal models. A disproportional increase in cellularity across thymocyte subsets, reduced relative proportions of cells at pre-DP TCR*αβ*^low^ stages, increased relative numbers of DP TCR*αβ*^low^ cells entering positive selection and their descendants, enhanced cell proliferation and reduced apoptosis in specific subsets, and the augmented thymic output of naïve T cells in Ovx rats likely reflected early disinhibition and changes in selection, possibly facilitating greater use of self-selection as an alternative mechanism for positive selection [39,42]. This imbalance in T cell subsets may contribute to a higher incidence of immune diseases, potentially linked to changes in the thymus before and after menopause. It is plausible that menopause-induced hormonal changes may result in thymic hypertrophy. Understanding these thymic changes could offer insights into thymic aging, improve immune function, decrease autoimmune diseases, and promote thymic regeneration.

### 2.4. Menstrual Cycle

The menstrual cycle is an important part of the female physiological cycle, but little is known about fluctuations in the number and function of immune cells during the menstrual cycle. Based on research findings, the Treg population undergoes expansion during the follicular phase of the menstrual cycle, attaining its peak prior to ovulation, while undergoing a significant decrease in the subsequent luteal phase [27]. The luteal phase is characterized by high progesterone/estrogen levels. Because of similarities between the luteal phase and pregnancy in terms of endocrine and immune factors, in the luteal phase the female body is described as being in a “pregnancy-like” state [43]. Compared to the early follicular phase, a reduction of CD4^+^ cells occurs in the luteal phase in accordance with the expected hormonal effects [44]. In contrast, a stimulatory effect of estrogens at periovulatory levels on IL-10 and interferon (IFN) -*γ* production has been described, while the effect on TNF-*α* appears biphasic, with lower concentrations stimulating its release (like that seen throughout the menstrual cycle) and elevated concentrations inhibiting it [45]. While progesterone induces IL-4 secretion, there is increased IL-4 production in the luteal phase of the cycle compared to the follicular phase [46]. At the same time, it has been found that the number of natural killer (NK) cells is increased in the luteal phase compared to the follicular phase [44]. These immune response genes such as *IL1-β*, nuclear factor-kappa B (*NF-κB*) *1*, signal transducer and activator of transcription (*STAT*) *3*, transforming growth factor-beta (*TGF-β*) were more frequently expressed during ovulation and the midluteal phase under estrogen/progestin regulation. IL-1*β* is the gene responsible for the proinflammatory response. NF-κB controls the expression of genes involved in immune development. In addition, TGF-*β* stimulates the differentiation of CD4^+^ T cells into Treg cells [47]. No study has found an effect of menstrual cycle on the thymus, but it has been found that IL-17 and IFN-*γ*, which represent helper T cell (Th) 17 and Th1, respectively, are significantly upregulated in a mouse model of polycystic ovary syndrome (PCOS). In contrast, IL-4, which represents the Th2 response, decreases to below normal levels, indicating that T cell polarization is involved in PCOS [48]. Perhaps these immune factors promote the process of CD4 differentiation into Treg cells and at the same time promote the potential of DP cells in the thymus to mediate delivery to SP. The dramatic hormonal fluctuations of the menstrual cycle that mimic pregnancy may drive the development of cTECs. What occurs in the thymus, the location for T cell development, merits further investigation.

### 2.5. Pathological Degeneration of the Thymus in Women

Degenerative thymus pairs increase susceptibility to various diseases, including lowered pathogen resistance, higher rates of autoimmune disorders, and impaired immune surveillance against tumors [49]. Autoimmune diseases affect 5–8% of the population, with women being more vulnerable than mean [50]. These diseases are a leading cause of disease-related mortality among women of reproductive age. Cervical, breast, and ovarian cancers are particularly threatening to women’s health.

#### 2.5.1. Systemic Lupus Erythematosus

Systemic lupus erythematosus (SLE) is characterized by the loss of tolerance to self-antigens, leading to the activation of autoreactive T cells. Sex hormones have been suggested to play a significant role in the pathogenesis of SLE [51]. It is more common in women and can be triggered by thymectomy. Structural changes observed in the thymus of SLE patients include reduced cortical thickness and abnormalities in the medulla. Lymphocytopenia is a significant feature, leading to atrophy and disorganization of the thymic medulla [52]. Elevated serum CXCL10 levels in SLE patients correlate with disease activity, and reduced thymus emigrants are linked to active disease [53]. Active SLE patients exhibit lower levels of T cell receptor excision circles (TRECs) in CD4^+^ T cells [54], indicating reduced TRECs in peripheral blood. Female SLE patients show increased expression of CXC receptor (CXCR) 3 [55] and TLR8 on naive T cells, leading to higher IFN-*α* production [56], which is crucial for SLE pathogenesis. Estrogen accelerates lupus progression in mice via ER-*α* signaling, increasing T cell activation, and expression of calcineurin and CD40L [57,58]. The absence of IL-23 in mice with lupus erythematosus affects mature thymocytes, particularly at the DP stage, leading to a reduction in the number of CD8 SP cells [59]. The loss of IL-23 also reduces the expression of IL-7 receptor, which has an important role in lymphocyte homeostasis. In mice with lupus, enhanced IL-7 signaling is associated with lymphocyte proliferation [59]. Additionally, deficiency of IL-23 led to a percentage decrease in mature CD44^+^CD62 L^+^ T cells both in the thymus and the periphery, disproportionately affecting CD8^+^ cells. SLE T cells overexpress CD44, which affects homing. IL-23 might play a significant role in the transport of lymphocytes to target tissues, which may be related to activation of STAT3, which enhances T cell adhesion and migration [59]. In mice with lupus erythematosus, the Th17 response is predominant, involving considerable accumulation of serum IL-17 and elevated phosphorylated STAT3 in CD4^+^ T cells. Related findings indicate the pathogenic role of unusual CD4^+^B220^+^ T cells in lupus in MRL/lpr mice based on their IL-17–producing capacity and conventional stimulatory function [60].

#### 2.5.2. Rheumatoid Arthritis

Rheumatoid arthritis (RA) is a systemic autoimmune disease that is prevalent among menopausal women. Rheumatoid arthritis, a rheumatic disease, is characterized by an imbalance in Th subsets, specifically a skewing towards Th1 and Th17 autoimmune responses at the expense of Th2 and Treg regulatory functions. In most patients, symptoms improve with pregnancy-induced acute thymic atrophy and the postpartum thymic remodeling process [51]. As RA progresses, age-related complications become more prevalent. Patients with RA demonstrate signs of premature immune senescence, including diminished thymic function, expansion of late-differentiated effector T cells, increased telomere depletion, and a pro-inflammatory phenotype. Early self-tolerance defects in RA are associated with immune system remodeling, primarily affecting T cells [61].

Thymus degeneration is a frequent occurrence in patients with RA, often accompanied by incomplete thymic enlargement followed by deficient thymic output, which is significantly associated with the proportions of CD4^+^ effector memory T cells, IL-7, Thymosin *β* (T*β*) 4, IL-9, Th9, and thymic stromal lymphopoietin (TSLP) [62,63,64,65]. The insufficient numbers of recent thymic emigrants (RTEs) could result from inadequate thymic T cell neogenesis, or alternatively could be a consequence of high CD4^+^ T cell turnover and homeostatic proliferation [66]. RA patients show a decreased frequency of TRECs in their peripheral blood mononuclear cells (PBMCs) [67]. Additionally, they present with lower levels of both CD4^+^ and CD8^+^ T cells, with a particularly significant reduction in CD4^+^ T cells [68]. There is also an expansion and accumulation of CD28^−^CD27^−^ T cells in RA patients. The prevalence of these cells tends to rise with age and is positively correlated with disease progression [69]. Moreover, the pool of naive CD4^+^ T cells in RA patients is approximately ten times smaller than that in age-matched controls, suggesting that they have lost around 90% of their available TCRs [70]. Thymic atrophy further restricts the expansion and production of naive T cells [71]. To preserve the numerical balance of the T cell compartment, compensatory mechanisms are activated, leading to homeostatic proliferation [72].

#### 2.5.3. Malignant Tumor

The “*International Cancer Statistics*” report revealed that cervical, endometrial, and ovarian cancers are diagnosed in 36.5%, 6.5%, and 12%, respectively, of individuals under the age of 45. Additionally [73], breast cancer is increasingly prevalent among young and middle-aged women [74]. A reduction in thymopoiesis is linked to a decline in immunosurveillance against cancer cells, potentially leading to alterations in the diversity or contraction of the TCR pool [75]. The pro-tumorigenic *γδ*-T cell subpopulation expressing V*γ*6 and *γδ*1 TCR chains has been associated with an elevated risk of developing cancer [76,77].

According to Zhou’s study, cervical cancer cells produce TSLP to facilitate angiogenesis and recruit eosinophils [78]. This downregulates the expression of microRNA (miR)-132 and promotes the proliferation and invasion of cervical cancer (CC) cells, ultimately contributing to the progression of CC [78]. Breast cancer patients exhibit significant abnormalities in thymic function, including impairment of thymic positive selection and the export of mature T cells to the peripheral blood [74]. Furthermore, tumorigenesis appears to impair stromal cell function, possibly due to tumor-derived factors. At advanced stages of tumorigenesis, immune function downregulation may be linked to thymus degeneration. Transgenic mice with lymphocyte-specific defects have shown NF-κB activation affects abnormal T cell development and positive selection of DP thymocytes [79,80].

Based on He et al.’s research, breast cancer patients exhibit cortical atrophy of the thymus, as observed in breast cancer model mice. These mice exhibited a reduction in thymic lymphocyte count, along with blurred connections between cortex and medulla regions leading towards disorganized structure compared with normal mice [74]. Cytokines secreted by breast cancer such as TSLP and TGF-*β* cause atrophy of the thymic cortex, which induces phenotypic changes of TECs, leading to reduced output from the organ [74]. Shi et al. found that changes in thymic function can impact both the development and metastasis of breast cancer by regulating Post-Translational Modification alpha (PTM*α*) and T*β*15 expression within the thymus, influencing tumor cell invasion [81].

Both physiological and pathological factors exert a combined influence on the alterations observed in the female thymus (Figure 2).

## 3. Female Sexual Hormones and Thymus

### 3.1. Estrogen and Its Receptors

Estrogen directly suppresses thymic progenitor cells in the bone marrow, such as Flt3^+^LSK cells, early thymic progenitors (ETPs), and DP cells [23]. This indicates that estrogen affects not only the thymus but also the proliferation of blood cell lineages, including potential thymic progenitors. Estrogen acts through various receptors, including nuclear receptors ER*α* and ER*β* and G protein-coupled receptor 30 (GPR30) [82]. These receptors have different roles: ER*α* blocks the development of CD44^+^CD25^−^ thymocytes, and it may influence the differentiation process of thymic epithelial cells through the Hedgehog pathway [82,83], while GPR30 induces apoptosis of TCR*β*^−/low^ DP thymocytes, potentially leading to thymus atrophy [82]. At the same time, estriol reduces the number of myeloid dendritic cells expressing receptors for thymic stromal lymphopoietin (CD11c^+^TSLP-R) and the inhibitory molecule B7-H3 (CD11cCD276) [35].

NF-κB signaling promotes survival and proliferation of *β*-selected thymocytes, while ER*α* inhibits NF-κB signaling in DN cells [82]. Estrogen downregulates Aire expression in mice via ER, thereby reducing the level of tissue-specific antigens (TSAs) involved in the tolerance process, and consequently affecting the function of mTECs [84]. Furthermore, estrogen may potentially interfere with the migration of thymocytes from the cortex to the medulla during the later stages of development by downregulating the expression of CCL25, CCL21, and CCL19 in mTECs [34]. It also induces thymus degeneration by inhibiting the TEC proliferative cycle, which may be related to ER*α*-specific regulation of miRs or Long non-coding RNA (lncRNA) s. The expression of sex-specific miRs during thymus degeneration is closely associated with the regulation of sex hormones. LncRNA-Gm12840 significantly enhances the viability and proliferation of MTEC1 induced by E2. Enhancing the viability and proliferation of MTEC1 cells causes significant upregulation of mRNA-*cdk1* and Wnt family member (*wnt*) *4*. In addition, it causes significant downregulation of mRNA-protein 53 (*p53*), *p21*, and *sfn* in the p53 signaling pathway, while *cyclinB1* is significantly upregulated [85]. MiR-16–5p induced via E2 affects MTEC1 by regulating *cyclinD1* and insulin-like growth factor-binding protein (*IGFBP*) *3* expression [86]. Estrogen may downregulate *cyclinB1* expression through the p53 signaling pathway and upregulate genes affecting MTEC1, such as *Gadd45* [87]. Moreover, Fas/FasL-mediated apoptosis is involved in thymic atrophy induction by E2 [88].

### 3.2. Progesterone

Though the role of progesterone in thymus degeneration appears to be insignificant, it exerts a significant immunological impact on the uterus during early pregnancy through the mediation of progesterone-induced blocking factor (PIBF) [89]. Both exogenous progesterone and elevated levels of endogenous progesterone during pregnancy effectively induce thymus degeneration.

The expression of C-C chemokine receptor (CCR) 9 and CXCR4 on TLPs and their ligands plays a role in the recruitment and migration of TLPs, while elevated progesterone levels downregulate the ligands’ expression, affecting homing and migration [34,90]. Elevated levels of progesterone can lead to widespread downregulation of CCL25, CXCL12, CCL21, and CCL19 expression, resulting in reduced homing of TLPs and impaired migration [34]. Progesterone significantly downregulates the expression of these factors in stromal cells, inhibiting DN cell proliferation as well as differentiation into DP cells [91]. Furthermore, progesterone may inhibit the expression of positive selection–related genes such as target gene–specific proteases for FoxN1 (*PSMB11*, *PRSS16*, *CTSL*) and stimulatory factors important for T cell receptor signaling (MHC-II, CD83) in cTECs [92,93,94], through PR signaling pathways which suppress the functional viability of TCRs during positive selection [95]. Furthermore, PR is predominantly localized in the cortex and expressed by FoxN1^+^ cTECs, with a significant upregulation observed during pregnancy, potentially synergizing with Wnt4 [37,96]. Klf4 maintains cTEC numbers by supporting cell survival and preventing epithelial-to-mesenchymal plasticity [97]. Then, mTECs express tissue-restricted antigens (TRA) to facilitate the differentiation of initial T cells into Tregs through dysregulated expression of both Aire and FEZ family zinc finger 2 (Fezf2), resulting in self-tolerance loss [98]. Progesterone enhances the negative selection of thymocytes by upregulating Aire expression, potentially promoting Treg differentiation and inducing immune tolerance towards the fetus [37]. Progesterone increases the expression of Fezf2 and improves the negative selection of thymocytes. This may promote differentiation of Treg cells and induce immune tolerance of the fetus. Fezf2 is expressed at high levels in mTEC cells, and mTECs with Fezf2^high^ express higher levels of Fezf2-dependent genes than Fezf2^low^ cells [98]. The presence of receptor activator of RANK on TECs facilitates ATI and Treg development during pregnancy and may also contribute to progesterone-induced enhancement of thymic negative selection function [99].

### 3.3. Luteinizing Hormone Releasing Hormone

Luteinizing hormone releasing hormone (LHRH), also known as gonadotropin-releasing hormone (GnRH), is also synthesized in small quantities within the fetal thymus. The immunomodulatory effects of LHRH include prevention of thymic atrophy, stimulation of T cell proliferation, activation of natural killer cells, and modulation of cytokine production. LHRH antagonists induce a decrease in the mass of the thymus and the number of mature T-lymphocytes in lymphoid organs and peripheral blood [100,101].

Marchetti et al. proposed that LHRH plays a major role as a signaling molecule in neuroendocrine–immune interactions [102]. Furthermore, injection of exogenous LHRH partially restored mitogen-dependent proliferation inhibition of thymocytes when both the hypothalamus and pituitary were simultaneously removed [103]. Sex hormones may also regulate synthesis of both thymosin and LHRH in the thymus and hypothalamus through interaction with specific receptors [104]. Velardi et al. found that IL-7 and DLL4 expression levels were significantly upregulated after treatment with LHRH-Ant, specifically affecting cTEC cells. Additionally, downstream Notch targets hes family bHLH transcription factor 1 (HES1), pre T cell antigen receptor alpha (PTCRA), and CD25 expression increased in developing T cells. This suggests that, within cTEC cells, LHRH-Ant can influence thymus development by modulating DLL4 expression within the Notch pathway [105]. LHRH also modulates thymic function through synthesis of the thymic cytokines IL-4, IL-10, IL-1*β*, IFN-*γ*, and TNF-*α*, which mediate the regulation of thymic development or morphogenesis [106].

### 3.4. Follistatin

Follistatin (FST), an inhibitor of activin A, plays crucial regulatory roles in the female reproductive tract. The significant increase in the proportion of mTEC^lo^ following puberty suggests a blockade in the differentiation from mTEC^lo^ to mTEC^hi^ [13]. Elevated production of FST can hinder activin A signaling, resulting in impaired differentiation of TEPC and mTEC [13]. The age-related rise in FST levels reduces the availability and signaling of activin A, which, combined with increased BMP (bone morphogenetic protein) 4 production, maintains TEPC and TEC precursors in an immature state, thereby disrupting the normal balance of differentiation. Ma et al. discovered that thymus size and thymocyte numbers were reduced in mice homozygous for follistatin-like 1 (FSTL1) gene deletion, leading to decreased proliferation of DN thymocytes and impaired T cell development [107]. In the thymus, mTEC cells are the primary cellular source of FSTL1. Therefore, lack of FSTL1 expression by mTEC cells results in reduced expression of inducible costimulators on activated T cell ligand (Icosl), which inhibits interactions between mTEC cells and CD4 SP thymocytes.

### 3.5. Prolactin

Developing thymocytes continuously produce prolactin (PRL), primarily derived from mTECs [108]. Expression of the prolactin receptor has also been detected on mTECs, and activation of this receptor induces proliferation of specific cell subsets [109]. Prolactin has demonstrated its significance in the survival and proliferation of early T cell progenitors, such as CD25^+^DN cells. Consequently, inhibition of prolactin expression impedes T cell development, resulting in accumulation of DN cells within the thymus [110]. Pre-treatment with PRL enhances the chemotactic effect of CXCL12 on total, DP, and SP thymocytes, while also influencing F-actin polymerization [111]. These findings suggest that prolactin contributes to physiological regulation of thymic function. Furthermore, it has been demonstrated that prolactin antagonizes the immunosuppressive effects mediated by glucocorticoids on the thymus under stressful conditions. Elevated circulating levels of prolactin protect thymocytes against glucocorticoid-induced apoptosis [112,113].

### 3.6. Growth Hormone

Growth hormone (GH) is produced by thymocytes, thymic epithelial cells, and peripheral lymphocytes [114,115]. GH production in the thymus can also be induced by the auxin-releasing peptide secreted by TEC and thymocytes [116]. GH has been demonstrated to stimulate T cell differentiation, proliferation, and cytokine production, and exhibits diverse activities in various cell types [117,118,119]. The age-related decline in circulating GH levels has been associated with thymus degeneration [120].

Our understanding of the effects of GH on the thymus can be traced back to the 1960s, when studies in Snell–Bagg mice revealed that these mice lacked growth cells and had a thymus that was only one-third of normal size in adulthood. GH therapy was utilized to reverse thymic atrophy by restoring microstructures and cytoarchitecture in the cortex, as well as promoting thymic DNA synthesis. GH exhibits pleiotropic effects on both the lymphoid and microenvironmental compartments of the thymus; while TEC produces GH, it also expresses GH receptors and proliferates in response to GH stimulation through upregulation of cytokinin A and cytokinin-dependent kinases [121]. Furthermore, GH modulates the thymic microenvironment by increasing levels of cytokines (IL-1*α*, IL-1*β*, and IL-6), chemokines such as CXCL12, and the thymic hormone thymosin [122].

In addition, Lee et al. [123] investigated mice with thymus degeneration and discovered that gastric starvation hormone inhibited the expression of pro-apoptotic proteins such as cysteine aspartate protease-3 (caspase-3) through promotion of GH secretion. This inhibition effectively prevented thymocyte apoptosis and enhanced TCR diversity. Lins et al. [124] observed that GH administration resulted in increased diversity of T cell receptors in thymocytes and facilitated the regeneration of TECs to restore immune function. Specifically, this included the production of extracellular matrix proteins and establishment of a more suitable thymic microenvironment, leading to a well-defined structure for cTECs and mTECs, ultimately supporting the development of mature T cells capable of performing immune functions.

### 3.7. Glucocorticoid

Systemic glucocorticoid (GC) is synthesized by the hypothalamic–pituitary–adrenocortical axis and binds to the glucocorticoid receptor (GR) upon secretion. The glucocorticoid receptor (GR) is expressed on various cell types throughout the body, including thymocytes [125]. Taves et al. demonstrated that, during thymic development, glucocorticoids synthesized by TECs specifically target CD4^+^CD8^+^TCR^high^ cells to promote positive selection [126]. The GR inhibits proliferation of thymocytes through binding with progesterone or PRs on membrane surfaces or via non-genomic pathways [37]. The involvement of TECs and thymocytes in the paracrine secretion of GC has been identified and confirmed, with immature DP thymocytes being the primary target cells for thymic glucocorticoid-associated immunosuppression [127]. Another study showed that GC influences differentiation and cytokine production within Tregs by promoting expansion of thymic Treg cells, increasing expression of forkhead box protein P3 (FoxP3) [128].

### 3.8. Androgen

Androgens (ARs) exert a potent influence on the thymus, which undergoes significant enlargement following castration and atrophy upon androgen treatment [129,130]. The thymic epithelium represents a target compartment for androgen, thereby impacting the generation of RTEs [131]. ARs suppress TEC expression of DLL4 and IL-7, both of which are crucial for DN survival and proliferation. Inhibition of androgen production through chemical means enhances cTEC DLL4 expression, as well as thymocyte expression of Notch target genes, resulting in maximal expansion of the DN, DP, and SP thymocyte populations [105]. CCL25 has been identified as playing a pivotal role in testosterone deficiency–induced alterations in thymopoiesis [132]. Wilhelmson observed an increased CCL25-positive area within the medulla, but not cortex, of epithelial cell-specific (E-ARKO) mice compared to control mice [131].

### 3.9. Others

It has been shown that thyrotropin-releasing hormone (TRH) receptor (TRHR)-related pathways are active in thymic cells and regulate thymic function [133]. In hyperthyroidism, the number of thymocytes is increased, leading to thymic hyperplasia [134]. The T3 nuclear receptor is expressed in both developing thymocytes and TECs [135]. Furthermore, systemic treatment with or intra-thymic injection of T3 increases thymocyte adhesion and migration to extracellular matrix molecules [136,137]. This effect is mediated by activation of the T3 nuclear receptor. In addition, mice lacking functional thyrotropin receptor expression have lower numbers of DP and SP thymocytes than wild-type mice, confirming the important role of Thyroid-stimulating hormone (TSH) in T cell development [138].

Leptin deficiency leads to premature thymic senescence accompanied by a severe depletion of thymocytes, which significantly impairs the differentiation of immature T cells into a more mature phenotype [139,140]. Specifically, leptin primarily promotes the differentiation of DP thymocytes into CD4^+^ single-positive mature thymocytes. Additionally, leptin protects thymocytes from apoptosis mainly through the regulation of key apoptosis-related proteins, such as BCL2, without affecting proliferation [139]. Leptin receptors are expressed in mTECs. Leptin can indirectly buffer DP thymocyte apoptosis and promote DN thymocyte proliferation in vivo, which leads to the restoration of thymic structure and increased TCR gene rearrangement. This attenuates systemic corticosteroid and proinflammatory cytokine responses, as well as intrathymic mechanisms orchestrated by medullary thymic epithelial cells and their soluble mediators, such as IL-7 [140].

## 4. Regulatory Mechanisms of Thymic Involution in Women

### 4.1. The Effects and Mechanisms of Female Sexual Hormones on Thymic Epithelial Cells

Female sex hormones can modulate thymic epithelial cells through the regulation of chemokines, miRs, lncRNAs, and other molecular mechanisms (Table 1 and Figure 3).

Progesterone may affect the gene encoding *FoxN1* in cTECs through PR signaling pathway and inhibit the survival rate of functional TCR thymocytes [92,93,94,95]. LHRH-Ant can influence thymus development by modulating DLL4 expression within cTEC cells in the Notch pathway [106].

Estrogen downregulates Aire expression in mice by binding to ER on mTECs [84]. The expression of sex-specific miRs such as miR-27b-3p, miR-16-5p, and miR-378a-3p is controlled by ER*α* through E2 in MTEC1 cells. The TGF-*β* signaling pathway, which is related to *IGFBP3*, may be an important signaling pathway by which miR-16-5p regulates cell proliferation. In addition, miR-27-3p, miR-140-3p, miR-378a-3p, miR-378d, and miR-186-5p are highly expressed not only in women, but also in young mice. In the future, further studies are required to explore the effects of these miRs on thymic epithelial cells [86,141]. LncRNA-Gm12840 significantly enhances MTEC1 induced in response to E2, upregulates mRNA-*cdk1* and *wnt4*, downregulates p53 signaling pathway components, and upregulates *cyclinB1* [85]. The downregulation of Aire and Fezf2 in mTECs affects Treg differentiation via progesterone, the function of Fezf2 depends on the lymphotoxin *β*-signaling axis, and RANK on TECs promotes sex hormone–mediated ATI and Treg development [9,98,99].

### 4.2. The Effects and Mechanisms of Female Sexual Hormones on Thymocytes

NF-κB signaling promotes survival and proliferation of *β*-selected thymocytes, while ER*α* inhibits NF-κB signaling in DN cells [82], blocking the DN1 to DN2 conversion and causing accumulation of DN1 cells and depletion of DP cells [50,82].

In contrast, CCR7 expressed on TLPs and CCL21 on mTEC^low^ and CCL19 on mTEC^high^ (both CCR7 ligands) may be involved in the migration of thymocytes from the cortex to the medulla [34]. Progesterone significantly downregulates the expression of these factors in stromal cells, inhibiting DN cell proliferation as well as differentiation into DP cells [91]. Progesterone can induce a decrease in Wnt4 levels, which in turn downregulates FoxN1 expression, leading to a decrease in the number and function of thymocytes [96]. Furthermore, PRL enhances the chemotactic effect of CXCL12 on total thymocytes, DP thymocytes, and SP thymocytes, while regulating F-actin polymerization [111].

## 5. Future Perspectives

During menopause, when female sex hormone levels decline, the thymus experiences a transient period of regeneration, leading to an increase in the size of the cortex [15]. The thymus undergoes involution at a slower rate in women compared to men. Women exhibit enhanced local interactions, specifically increased cell–cell interactions between cTECs and class II^+^ DP thymocytes, as well as within the positive selection niche that includes DCs and DPs. These findings suggest that the proportionally larger female thymic cortex may facilitate increased cross-presentation by DCs and cTECs to class II^+^ DPs, potentially enhancing self-selection [21]. This process may be further reinforced during menopause. Additionally, the expression of cTEC-related chemokines, including DLL4, FoxN1, Psmb11, Ctsl, and CCL25, which are regulated by estrogen and progesterone, is significantly higher in women than in men [12]. These chemokines play a crucial role in postpartum regeneration and are implicated in the activation, proliferation, and differentiation of T cells [36]. During menstruation, women undergo hormonal fluctuations akin to those observed during pregnancy. Could such hormonal variations potentially stimulate the regenerative capacity of cTECs? This phenomenon may elucidate why thymic involution occurs more gradually in women compared to men. However, the precise underlying mechanism warrants further in-depth investigation.

A study found that the cortical response observed in the amplification of cortical areas does not directly stem from cTECs, but instead reflects the responses of medullary stroma to the endocrine environment, which subsequently leads to the regulation of cTEC morphology [142]. After ovariectomy, both the number of DP thymocytes and the number of mature single-positive thymocytes increases [42], indicating that the mechanism of thymus regeneration during menopause may also enhance negative selection, which in turn negatively regulates the morphology and population of cTECs. Research has found that estrogen can significantly reduce the expression of Aire, which is involved in the differentiation and maturation of mTECs and the production of chemokines [84]. Overexpression of miR-16-5p induced by E2 inhibits the proliferation of MTEC1 cells by downregulating *cyclinD1* and *IGFBP3*. The IGFBP3-related TGF-*β* signaling pathway may be an important signaling pathway by which miR-16-5p regulates cell proliferation [86]. Aire deficiency reduces the number of *β*-gal^+^ epithelial cells and enhances thymic output [143]. This may induce apoptosis of mTECs and favor the cross-presentation of TSA proteins present in apoptotic cells to thymic dendritic cells, suggesting that thymocyte self-selection may play a large role during positive selection.

Resveratrol and curcumin, both of which act as natural estrogen inhibitors, have been shown to enhance thymus function [144,145]. Resveratrol can upregulate the expression of FoxN1, promote the generation of cTEC, and thereby facilitate thymus regeneration [144]. On the other hand, curcumin significantly enhances the expression level of Aire, and its mechanism may be related to promoting negative selection in the thymus [145]. Additionally, vitamin D downregulates the expression of estrogen receptor alpha and is essential for normal Aire^+^ TEC maturation, TRA gene expression, and promoting thymic tissue development [146,147].

## 6. Conclusions

The thymus is a central immunological organ that undergoes gradual involution and atrophy with advancing age. The degenerative process of the female thymus is influenced by multiple factors, including gender-specific diseases, pregnancy, menopause, and other contributing factors. Women demonstrate a comparatively slower rate of thymus degeneration in comparison to men, likely attributable to their heightened potential for cTEC renewal acquired during evolution and multiple hormonal regulations in vivo. Menopause, an inevitable facet of female aging, which involves the rapid decline of hormones within the body, may exert some influence on thymus function. Further evidence is required to substantiate this notion. Perhaps we can find the answer by constructing hormone–TEC–chemokines and using single-cell omics technology. This will augment our understanding of the regulation of thymus degeneration in women.

## Figures and Tables

**Figure 1 ijms-26-03014-f001:**
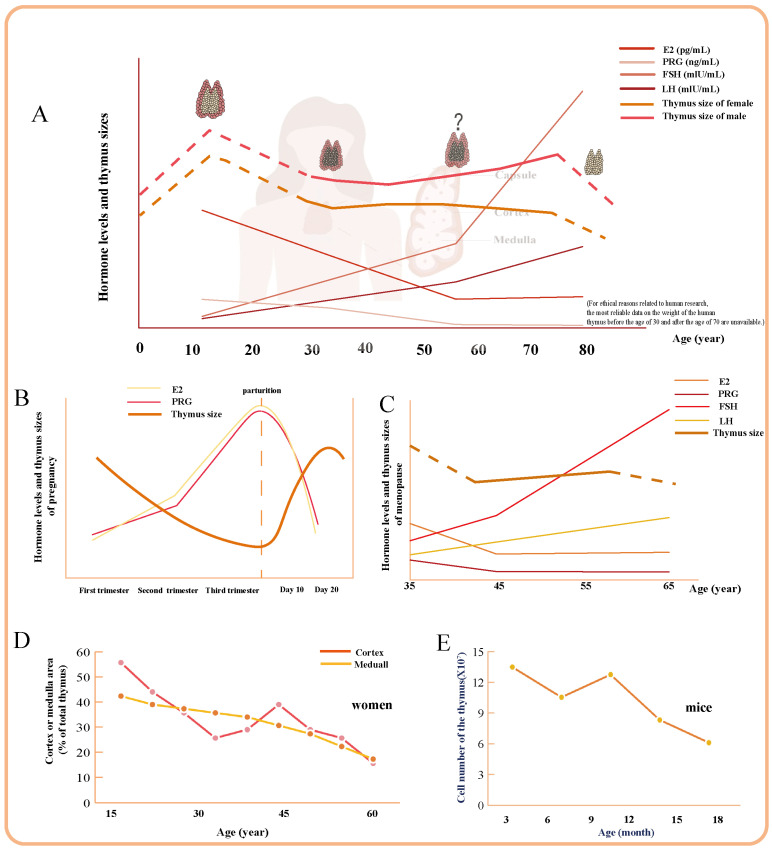
The human and mouse thymus across the life span. (**A**) Hormone levels and thymus sizes. (**B**) Hormone levels and thymus sizes in pregnancy. (**C**) Hormone levels and thymus sizes in menopause. (**D**) Cortex or medulla area in women. (**E**) Cell numbers in the mouse thymus. E2, estradiol; PRG, progesterone; FSH, follicle-stimulating hormone; LH, luteinizing hormone. (Both before age 30 and after age 70 are represented by dashed lines because values are estimated; after 69 there is a survivor bias in the thymus of men, resulting in another increase in thymic volume at this stage.

**Figure 2 ijms-26-03014-f002:**
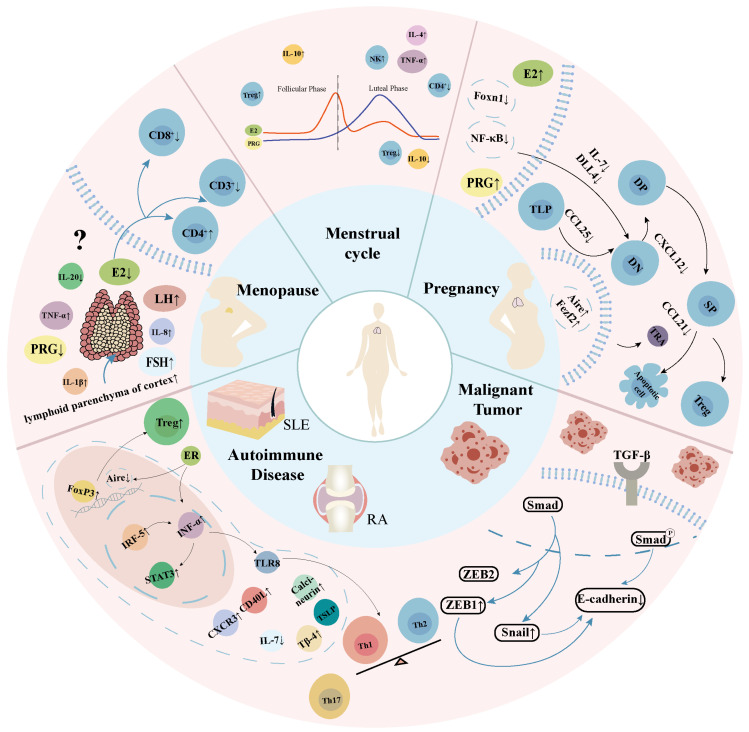
Physiologic and pathological thymus degeneration in women. E2, estradiol; PRG, progesterone; FSH, follicle-stimulating hormone; LH, luteinizing hormone; ER, estrogen receptor; SLE, systemic lupus erythematosus; RA, rheumatoid arthritis; IL-20, interleukin-20; IL-8, interleukin-8; IL-4, interleukin-4; IL-10, interleukin-10; NK, natural killer; IL-1*β*, interleukin-1 beta; TNF-*α*, tumor necrosis factor-alpha; FoxN1, forkhead box N1; NF-κB, nuclear factor-kappa B; DP, CD4^+^CD8^+^ double-positive; DN, CD4^−^CD8^−^ double-negative; SP, single-positive; Treg, regulatory T cell; TLP, T lymphocyte progenitor; IL-7, interleukin-7; DLL4, *δ*-like ligand 4; CCL25, chemokine (C-C motif) ligand 25; CXCL12, chemokine (C-X-C motif) ligand 12; CCL21, chemokine (C-C motif) ligand 21; TRA, tissue restricted antigens; Aire, autoimmune regulator; Fezf2, FEZ family zinc finger 2; FoxP3, Forkhead box protein P3; IRF-5, interferon regulatory factor-5; INF-*α*, interferon-alpha; STAT3, signal transducer and activator of transcription 3; TLR8, Toll-like receptor 8; TSLP, thymic stromal lymphopoietin; CXCR3, CXC receptor 3; CD40L, CD40 ligand; Th, helper T cell; ZEB1, zinc finger E-box binding homeobox 1; ZEB2, zinc finger E-box binding homeobox 2; Smad, suppressor of mother against decapentaplegic; Snail, snail family transcriptional repressor 1; TGF-*β*, transforming growth factor-beta.

**Figure 3 ijms-26-03014-f003:**
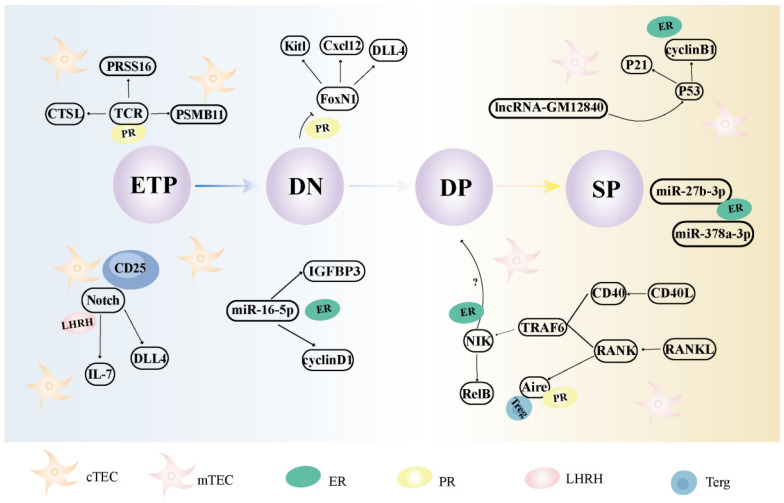
The influence of female sexual hormones on thymic epithelial cells. ER, estrogen receptor; PR, progesterone receptor; LHRH, Luteinizing hormone releasing hormone; cTEC, cortical thymic epithelial cell; mTEC, medullary thymic epithelial cell; Treg, regulatory T cell; IL, interleukin; FoxN1, forkhead box N1; NF-κB, nuclear factor-kappa B; DP, CD4^+^CD8^+^ double-positive; DN, CD4^−^CD8^−^ double-negative; SP, single-positive; DLL4, *δ*-like ligand 4; CCL, chemokine (C-C motif) ligand; CXCL, chemokine (C-X-C motif) ligand; Aire, autoimmune regulator; IGFBP, insulin-like growth factor-binding protein; NIK, NF-κB–inducing kinase; RANK, receptor activator of nuclear factor-κB; RANKL, receptor activator of NF-κB ligand; TRAF6, TNF receptor associated factor 6; CD40L, CD40 ligand; TTCR, cell receptor; CTSL, cathepsin L; PSMB, proteasome subunit beta; PRSS, serine protease.

**Table 1 ijms-26-03014-t001:** The effects of female sexual hormones on thymic epithelial cells.

Hormone	Cell Type	Action	Reference
Estrogen	mTEC	Reduces the expression of Aire in mTECs and TRAs and inhibits T cell negative selection; induces overexpressing of miR-16-5p and downregulates cyclinB1 and IGFBP3 in mTECs	[82,84,86]
Progesterone	cTEC mTEC	Influences Klf4 to maintain cTEC numbers; affects the expression of FoxN1, potentially synergizing with Wnt4; upregulates Aire expression in mTECs	[37,96,97]
Luteinizing hormone releasing hormone	cTEC	ModulatesDLL4 expression in the Notch pathway	[105]
Follistatin	mTEC	Inhibits activin A signaling, resulting in impaired TEPC and mTEC differentiation	[13]
Prolactin	mTEC	Enhances the chemotactic effect of CXCL12, while also influencing F-actin polymerization in mTECs	[111]
Growth hormone	cTEC, mTEC	TECs are regulated by increased cytokines (IL-1*α*, IL-1*β*, and IL-6) and secretion of the chemokine CXCL12, as well as the thymus hormone thymosin	[122]
Glucocorticoid	mTEC	Targets CD4^+^CD8^+^TCR^high^ cells to promote positive selection	[126]
Thyroid hormone	cTEC mTEC	Affects the expression of DP and SP cell subsets	[138]
Androgen	cTEC	Suppresses cTEC expression of DLL4 and IL-7; influences thymocyte expression of Notch target genes	[105]
Leptin	mTEC	Promotes the proliferation of DN thymocytes; regulates IL-7 in mTECs	[140]

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
