# Peer review of "Thymus Degeneration in Women and the Influence of Female Sexual Hormones on Thymic Epithelial Cells"

_ijms, 2025, doi:10.3390/ijms26073014_

Round 1

Reviewer 1 Report

Comments and Suggestions for Authors

The manuscript (review article) that entitled by “Thymus degeneration in females and the influence of female sexual hormone on thymic epithelial cells”, discussed the factors affecting the degenerative process of the female thymus, including gender-specific diseases, pregnancy, menopause, and other contributing factors. Their findings indicate that females demonstrate a comparatively slower rate of thymus degeneration in comparison to males, likely attributed to their heightened potential for cortical thymic epithelial cell (cTEC)  renewal acquired during evolution and multiple hormonal regulations in vivo. Also, menopause can induce immune dysfunction due to the rapid decline of hormones within the body, which may exert some influence on thymus function. The manuscript is generally well-addressed and well-cited; However, I have some comments and suggestions.

Line 12: All above mentioned at the abstract is not focused directly to the conducted study. Please rewrite to be more focused. It is required to place the question addressed in a broad context and highlight the purpose of the study.

Line 19: Please mention these regulation principles and summarize the article's main findings. The abstract should be an objective representation of the article: it must not contain results which are not presented and substantiated in the main text and should not exaggerate the main conclusions.

Line the same reference number 3 was mentioned 2 times subsequently. Please try to integrate all in one part for the same reference.

Line 97: Please try to focus on using the literature for thymus in female human being not in mice to be more focus and concentrated. The difference between the species plays an important role for getting different results.

Line 110: as mentioned " This sexual dimorphism " please revise this paragraph and its relationship for the previously mentioned and the further mentioned paragraphs to keep the sequence of the literature  in a good shape.

Line 130: Please revise any research studies did ovariectomy that mentioned at the literature. There is a difference in hormonal levels during the ovariectomy and the menopause.

Line 191: The same repeated words mentioned again by the same reference (27) mentioned before at line 130.

Line 207: Please recheck if you still need to mention the menstrual cycle paragraph as already know.

Line 227: I think it will better if you moved this section of " Pathological degeneration of thymus in females " to be AFTER not before point number 3. Female sexual hormones and thymus.

Line 339: Please revise each hormones. There are some details away from their effect on thymus and thymic cells.

Line 552: I suggest to this part "Regulatory mechanisms of thymic involution in female " as a separate main part not subtitle.

References number 4, 8, 85 and 144 is incomplete. Upon journal guidelines for journal articles, it should have Abbreviated Journal Name YearVolume and page range. Please revise.

Comments on the Quality of English Language

Major editing of English language required.

Author Response

Reviewer #1: The manuscript (review article) that entitled by “Thymus degeneration in females and the influence of female sexual hormone on thymic epithelial cells”, discussed the factors affecting the degenerative process of the female thymus, including gender-specific diseases, pregnancy, menopause, and other contributing factors. Their findings indicate that females demonstrate a comparatively slower rate of thymus degeneration in comparison to males, likely attributed to their heightened potential for cortical thymic epithelial cell (cTEC)  renewal acquired during evolution and multiple hormonal regulations in vivo. Also, menopause can induce immune dysfunction due to the rapid decline of hormones within the body, which may exert some influence on thymus function. The manuscript is generally well-addressed and well-cited; However, I have some comments and suggestions.

Point 1&2: Line 12: All above mentioned at the abstract is not focused directly to the conducted study. Please rewrite to be more focused. It is required to place the question addressed in a broad context and highlight the purpose of the study.Line 19: Please mention these regulation principles and summarize the article's main findings. The abstract should be an objective representation of the article: it must not contain results which are not presented and substantiated in the main text and should not exaggerate the main conclusions.
Response 1&2:Thank you for your valuable advice. We have revised the abstract to ensure it provides a more objective and accurate representation of the article.

Point 3:Line the same reference number 3 was mentioned 2 times subsequently. Please try to integrate all in one part for the same reference.
Response 3: Thank you for your advice. We have integrated the contents of the reference number 3.

Point 4: Line 97: Please try to focus on using the literature for thymus in female human being not in mice to be more focus and concentrated. The difference between the species plays an important role for getting different results. 

Response 4: Thank you for your advice. Despite the previous differences between mice and humans, due to ethical reasons, direct research on the human thymus is rare, but we have supplemented the content of the article to suggest that there are similarities between mice and humans. For example, "Additionally, female thymic lobes contain significantly higher numbers of double-positive cells (DPs) and cTECs [21]. In  eight-week-old mice, female mice have slower medulla atrophy compared to males, with more DP cells but smaller double negative (DN) and single positive (SP) cells than males [22].".

Point 5: Line 110: as mentioned " This sexual dimorphism " please revise this paragraph and its relationship for the previously mentioned and the further mentioned paragraphs to keep the sequence of the literature  in a good shape.
Response 5: Thank you for your insightful advice. We appreciate your attention to the logical flow of the manuscript. As suggested, we have carefully revised the paragraph to strengthen its connection to the preceding and subsequent paragraphs. Specifically, "Women exhibit a slower rate of thymus involution between the ages of 20 and 30 years than men [14]. This sexual dimorphism in thymus attenuation is likely attributable to variations in the hormonal environment. The low expression of APOD, a gene associated with hormone signaling, in female fibroblast populations suggests an earlier onset of thymic involution in males relative to females [21]. ".

Point 6: Line 130: Please revise any research studies did ovariectomy that mentioned at the literature. There is a difference in hormonal levels during the ovariectomy and the menopause.

Response 6: Thank you for your valuable advice. We agree that there are some differences between menopausal and ovariectomy hormone levels, particularly in terms of the abrupt decline in estrogen and progesterone post-surgery compared to the gradual hormonal fluctuations in menopause. However, both conditions share similarities in immune systems, such as CD4/CD8 ratio in peripheral blood. At the same time, the thymus showed an increase in cortical areas after menopause and ovariectomy, suggesting that similar mechanisms may be behind it.

Point 7: Line 191: The same repeated words mentioned again by the same reference (27) mentioned before at line 130.

Response 7: Thank you for your attentive advice regarding the repeated use of Reference 27 in Line 130. We have made some changes to it, our intent was to emphasize the shared thymic changes, increased thymus weight and cortical expansion observed in both menopausal and ovariectomized models.

Point 8: Line 207: Please recheck if you still need to mention the menstrual cycle paragraph as already know.

Response 8: Thank you for raising this important point. While the menstrual cycle’s hormonal fluctuations (e.g., estrogen, progesterone) and associated immune modulators (e.g., IL-10, IFN-γ, TNF-α) are well-documented, we discoverd that direct evidence linking these cyclical changes to thymic function remains limited in current literature.We also added a little bit of content to side speculate on possible effects on the thymus. By acknowledging this gap, we hope to encourage research into how cyclical immune adaptations in females—often overlooked in immunology—might predispose to long-term health outcomes. We are happy to shorten or remove this section if the reviewer deems it tangential, but we believe it raises a novel question pertinent to female-specific immunology. Thank you for your consideration.

Point 9: Line 227: I think it will better if you moved this section of " Pathological degeneration of thymus in females " to be AFTER not before point number 3. Female sexual hormones and thymus.

Response 9: Thank you for your thoughtful suggestion regarding the placement of the section "Pathological degeneration of thymus in females". We appreciate your effort to improve the manuscript’ s logical flow. We used this order because before "Pathological degeneration of thymus in females", menopause and pregnancy, is Physiologic thymus degeneration in females. We arranged this way to better illustrate the effect of physiological and pathological changes in women on the thymus.

Point 10: Line 339: Please revise each hormones. There are some details away from their effect on thymus and thymic cells.

Response 10: Thank you for your meticulous feedback regarding the hormone-related details. We sincerely appreciate your attention to the precision and focus of our discussion. As suggested, we have carefully revised this section to eliminate extraneous details and ensure that the content strictly emphasizes the effects of each hormone on thymic structure and function, particularly their roles in thymocyte development, epithelial cell regulation.  Modifications have been highlighted in red in the article.

Point 11: Line 552: I suggest to this part "Regulatory mechanisms of thymic involution in female " as a separate main part not subtitle.

Response 11: Thank you for your valuable suggestion regarding the structural organization of the manuscript. We greatly appreciate your insight into enhancing the clarity and logical flow of the text.  As recommended, we have revised the section titled "Regulatory Mechanisms of Thymic Involution in Females" by promoting it from a subtitle to a standalone main section (now Section 4).

Point 12: Line 552: References number 4, 8, 85 and 144 is incomplete. Upon journal guidelines for journal articles, it should have Abbreviated Journal Name Year, Volume and page range. Please revise.

Response 12: Thank you for your meticulous attention to the formatting details of the references. We sincerely appreciate your guidance in ensuring compliance with the journal’s guidelines. We have carefully revised References to include the abbreviated journal names, publication years, volumes, and page ranges as required.

However, for some documents, we did not find such as [4] and [8] Volume and page range. I am really sorry. Regarding the master’s theses cited in the manuscript (e.g., Reference [85] CN. W. Effect of 17B-Estradiol on Expression of LncRNA in Mouse Thymic Epithelial Cells. MA thesis. South China Agricultural University. Guangdong, China, 2018.), we acknowledge that these sources inherently lack page ranges.  To address this, we have explicitly labeled them as "Master’ s Thesis" in the reference list to clarify their format and avoid ambiguity. We regret any oversight in the initial submission and have thoroughly cross-checked all references to align with the journal’s standards.

Reviewer 2 Report

Comments and Suggestions for Authors

The manuscript provides a comprehensive review of thymus degeneration in females and hormonal influences but lacks groundbreaking insights. Several related reviews have been published in that regard that covered hormonal regulation of thymic epithelial cells (TECs). In general, the manuscript consolidates existing knowledge but still lacks unique novelty for their solutions discussion. I suggest discussing the integrated multi-omics data to identify novel hormone-TEC regulatory networks. And, discussing the proposed therapeutic targeting of hormone receptors (e.g., ERα antagonists) using in vivo models. Besides, explore dietary phytoestrogens (e.g., genistein) as modulators of thymic function. A discussion of recent advancements (e.g., single-cell RNA sequencing in TECs, and CRISPR-based studies) should be provided. I suggest the author consider the interesting related study to the single-cell RNA sequencing in metabolic transcriptomic visualization PMID: 38896978 DOI: 10.1016/j.bios.2024.116504. Abstract has several phrases to be reconsidered carefully, like “result an obvious sexual dimorphism”. Links to hormonal fluctuations to autoimmune susceptibility and thymic aging should be more clarified by highlighting the sexual dimorphism in immune disorders comprehensively because of the lack of critical analysis that could make conflicting evidence (e.g., estrogen’s dual role in thymic atrophy vs. T cell output).  Figure 1 and 2 descriptions lack clarity (e.g., “dashed lines” in Figure 1 need explanation). Table 1. Should be checked for its typos. Page 2,  “Polyhormones” should be clarified. Table 1 formatting inconsistencies (e.g., missing hormone actions for some rows). In addition, I suggest mentioning more about the bulk RNA-seq datasets discussion to predict hormone-responsive genes in TECs. Please clarify more about how the targeted therapies to delay thymic degeneration in aging women could be more advanced, including the current mechanisms. Besides, the mentioned limited data on hormone receptor crosstalk in TECs. The role of non-coding RNAs (miR-27b-3p) in thymic aging in more deepening characteristics. In conclusion, major revisions are required to emphasize innovation (e.g., therapeutic hypotheses novel insights).

Comments on the Quality of English Language

The English could be improved to more clearly express the research.

Author Response

Reviewer #2: The manuscript provides a comprehensive review of thymus degeneration in females and hormonal influences but lacks groundbreaking insights. Several related reviews have been published in that regard that covered hormonal regulation of thymic epithelial cells (TECs). In general, the manuscript consolidates existing knowledge but still lacks unique novelty for their solutions discussion. 

Point 1&2: I suggest discussing the integrated multi-omics data to identify novel hormone-TEC regulatory networks. And, discussing the proposed therapeutic targeting of hormone receptors (e.g., ERα antagonists) using in vivo models. Besides, explore dietary phytoestrogens (e.g., genistein) as modulators of thymic function. A discussion of recent advancements (e.g., single-cell RNA sequencing in TECs, and CRISPR-based studies) should be provided. I suggest the author consider the interesting related study to the single-cell RNA sequencing in metabolic transcriptomic visualization PMID: 38896978 DOI: 10.1016/j.bios.2024.116504. In addition, I suggest mentioning more about the bulk RNA-seq datasets discussion to predict hormone-responsive genes in TECs. Please clarify more about how the targeted therapies to delay thymic degeneration in aging women could be more advanced, including the current mechanisms.

Respons1&2: Thank you for your insightful and comprehensive suggestions, which have greatly enriched the depth and translational relevance of our manuscript. We deeply appreciate your expertise in guiding us toward integrating cutting-edge methodologies and therapeutic perspectives. In response to your feedback, we have made the following revisions: We supplemented the effects of ERα antagonist vitamin D and natural estrogen inhibitors resveratrol and curcumin on the thymus, and explored the possible mechanisms. At the same time, we increased the transcriptome and single-cell omics analysis of gender differences, and found that women had stronger expression of CTEC-related factors. We analyzed some differences between sexsex miRNA and hormone action on miRNA. Female individuals exhibit enhanced local interactions, specifically increased cell-cell interactions between cTECs and DP thymocytes. This may become a potential mechanism to delay thymic degeneration in aged women. However, as this manuscript is a review, there are few articles on monocytomics at present, but this is an innovative point, worthy of further research.

Point 3: Abstract has several phrases to be reconsidered carefully, like“result an obvious sexual dimorphism”.
Response 3: We sincerely thank you for your invaluable guidance and insightful critique. In response to your feedback, we have revised the abstract to strengthen clarity to eliminate ambiguity and ensure alignment with the study’s core findings.

Point 4: Figure 1 and 2 descriptions lack clarity (e.g., “dashed lines” in Figure 1 need explanation). 

Response 4: Thank you for your valuable comments on the clarity of Figure 1 and 2. We sincerely appreciate your great interest in detail, which greatly improved the accuracy of our draft. In response to your concerns, we illustrate the meaning of the dashed line In the figure legends. Pointing out that this is an estimate of thymus size. In addition, we make a brief clarification in the main text (for ethical reasons related to human studies, the most reliable data on human thymic weight are not available between the ages of 30 and 70). Therefore, the relevant sections in Figure 1a are presented intermittently for reference purposes only).

Point 5: Table 1. Should be checked for its typos. Page 2,  “Polyhormones” should be clarified. Table 1 formatting inconsistencies (e.g., missing hormone actions for some rows).

Response 5: Thank you for your meticulous attention to detail in reviewing Table 1. We sincerely appreciate your feedback.  Based on your suggestions, we have thoroughly revised the table.

Point 6: Besides, the mentioned limited data on hormone receptor crosstalk in TECs. The role of non-coding RNAs (miR-27b-3p) in thymic aging in more deepening characteristics.

Response 6: Thank you for your insightful suggestions regarding the role of non-coding RNAs in thymic aging and the need to deepen the characterization of hormone receptor crosstalk in cTECs. We sincerely appreciate your expertise and we added some factors that are related to CTEC and highly expressed in women. Moreover, Since there are few articles on the effects between the thymus and miR-27b-3p, miR-16-5p is more closely associated with E2 and mTEC, so we supplemented the mechanism of miR-16-5p.

Reviewer 3 Report

Comments and Suggestions for Authors
  1. The regulation of thymus by LHRH and leptin, are described briefly and lack direct experimental evidence to support them. It is recommended to supplement relevant research data or cite more specific literature.
  2. More clinical or animal model data is needed to validate the hypothesis of "menopausal thymic hypertrophy", such as transient hyperplasia caused by a sudden drop in estrogen.
  3. In Figure 1, the age range indicated by the dashed line, have not been fully explained in the main text, which can easily cause confusion for readers. Please correct it.
  4. The classification of the "Effects of Hormones on Thymocytes" section in Table 1 is not clear, such as thyroid hormones, GnRH, etc. under "Other Hormones", and it is recommended to reclassify them according to function or signaling pathway. Please correct it.
  5. Some conclusions rely on indirect evidence, such as "speculation of menopausal thymic hypertrophy", and direct experiments, including histological analysis of thymus in menopausal women need to be supplemented. And, here is limited quantitative data on the dynamic changes in hormone levels, such as Treg fluctuations during the menstrual cycle. It is recommended to include the correlation analysis between hormone concentration and immune indicators.
  6. Some key mechanisms, such as progesterone regulating FoxN1 through Wnt4, have outdated references. It is recommended to supplement the latest research support.
  7. Some sentences are lengthy, such as the description of estrogen receptors in section 3.1.1, it is recommended to split them to improve readability. In addition, a few grammar errors, such as "result an obvious sexual dimorphism" should be changed to "resolving in", require full proofreading.

Major Revision: The paper requires significant improvements in language, statistical interpretation, and discussion depth before it can be considered for publication.

Comments on the Quality of English Language

The writing contains numerous grammatical and syntactical errors, making some sections difficult to read. The manuscript should undergo thorough proofreading or professional language editing.

Author Response

Point 1: The regulation of thymus by LHRH and leptin, are described briefly and lack direct experimental evidence to support them. It is recommended to supplement relevant research data or cite more specific literature.

Response 1: Thank you for your constructive feedback on the sections discussing the regulation of the thymus by LHRH (GnRH) and leptin. We deeply appreciate your guidance in strengthening the scientific rigor of these discussions. As suggested, we have revised the manuscript to include direct experimental evidence and additional citations to support these mechanisms. The updates are summarized below: Luteinising hormone releasing hormone (LHRH) also known as Gonadotropin-releasing hormone (GnRH) is also synthesised in small quantities within the foetal thymus. Immunomodulatory effects of LHRH include prevention of thymic atrophy, stimulation of T cell proliferation, activation of natural killer cells, and modulation of cytokine production;Leptin deficiency leads to premature thymic senescence accompanied by a severe depletion of thymocytes, which significantly impairs the differentiation of immature T cells into a more mature phenotype.

Point 2&5: More clinical or animal model data is needed to validate the hypothesis of "menopausal thymic hypertrophy", such as transient hyperplasia caused by a sudden drop in estrogen. Some conclusions rely on indirect evidence, such as "speculation of menopausal thymic hypertrophy", and direct experiments, including histological analysis of thymus in menopausal women need to be supplemented. And, here is limited quantitative data on the dynamic changes in hormone levels, such as Treg fluctuations during the menstrual cycle. It is recommended to include the correlation analysis between hormone concentration and immune indicators.

Response 2&5: Thank you for your insightful suggestion. We fully agree that additional clinical or experimental data would enhance the mechanistic understanding of this phenomenon. We have revised Section "2.3 "Special Modifications in the Thymus of Postmenopausal Women" to address this concern. Clinical Data Limitations: We explicitly acknowledge the scarcity of direct clinical data on thymic changes in menopausal women, largely due to ethical constraints in obtaining longitudinal thymic tissue samples and historical under representation of female-specific aging studies. While direct human data remain limited, the inclusion of ovariectomy model findings and indirect clinical correlations provides a robust foundation for the hypothesis. Hormonal fluctuations during the menstrual cycle, which we have increased,and changes in Terg, which may be associated with IL-10, IFN-γ,TNF-α,and IL-4, are also described. We sincerely appreciate your critique, which has prompted a more nuanced and evidence-based discussion of this topic. Please let us know if further clarifications or expansions would improve the manuscript’ s rigor.

Point 3: In Figure 1, the age range indicated by the dashed line, have not been fully explained in the main text, which can easily cause confusion for readers. Please correct it.

Response 3: Thank you for your valuable feedback regarding the clarity of Figure 1. We sincerely appreciate your meticulous  attention to detail, which has greatly improved the rigor of our manuscript. In response to your concern, we have revised the figure legend to explicitly explain the dashed line,  indicating it represents an estimate of thymus size. Additionally, we added a brief clarification in the main text (For ethical reasons related to human research, the most reliable data on the weight of the human thymus before the age of 30 and after the age of 70 are unavailable. Consequently, the relevant sections in Figure 1A are indicated by dashed lines for reference purposes only.).

Point 4: The classification of the "Effects of Hormones on Thymocytes" section in Table 1 is not clear, such as thyroid hormones, GnRH, etc. under "Other Hormones", and it is recommended to reclassify them according to function or signaling pathway. Please correct it. 

Response 4: Thank you for your insightful feedback on the organization of Table 1. We greatly appreciate your effort to enhance the clarity. As suggested, we have revised the table to improve its classification and functional coherence. Since Gonadotropin-Releasing Hormone (GnRH) and Luteinizing Hormone-Releasing Hormone (LHRH) represent the same class of molecules, we consolidated them under a single entry to avoid redundancy and improve clarity. We are grateful for your meticulous critique, which has significantly strengthened the manuscript’ s rigor.

Point 6: Some key mechanisms, such as progesterone regulating FoxN1 through Wnt4, have outdated references. It is recommended to supplement the latest research support.

Response 6: Thank you for your valuable feedback regarding the need to update references for key mechanisms such as progesterone-mediated regulation of FoxN1 through Wnt4. We sincerely appreciate your guidance in ensuring the manuscript reflects the latest advancements in the field. In response to your suggestion, we have revised the relevant section to incorporate recent studies that strengthen and expand upon the proposed mechanisms. For example: Klf4 maintains cTEC numbers by supporting cell survival and preventing epithelial-to-mesenchymal plasticity.

Point 7: Some sentences are lengthy, such as the description of estrogen receptors in section 3.1.1, it is recommended to split them to improve readability. In addition, a few grammar errors, such as "result an obvious sexual dimorphism" should be changed to "resolving in", require full proofreading.

Response 7: Thank you for your meticulous feedback on improving the readability and grammatical precision of the manuscript. We sincerely appreciate your efforts to enhance the clarity and rigor of our work.  We have revised it to improve the readability of the article.

Reviewer 4 Report

Comments and Suggestions for Authors

Dear Authors,

Commonly, an interesting manuscript with some new aspects for the fiels.

However, I have some minor and some major abjections for it:

Minor:

1) Title. Please, add here someting about "historical aspect" too, as you have 25 (!) previous century sources. You cant leave them without explanation, but if you will change slightly the title, this can work!

2) add the plan of the review at the end of Introduction; also data bases used, key words, inclusion criteria for the papers, time period for this research;

3) indicate, pelase, for the Fig. 3, from which source this picture is adopted;

Major objection

! thymus is developed from 6 different types of cells. You cant explain the involution without analysis whats going on with each of these types in thymus. Thus, please, include a subsection with this description. Whats going on? In men, female, are there differences? Which cells suffers the most?  From which cells does the involution start? whats happening with this involution in ontogenetic aspect (during female life) etc? Thank you. 

Author Response

Point 1: Title. Please, add here someting about "historical aspect" too, as you have 25 (!) previous century sources. You cant leave them without explanation, but if you will change slightly the title, this can work!

Response 1: Thank you for your thoughtful suggestion regarding the title and the historical context of the cited literature.  We  deeply appreciate your guidance in refining the manuscript’ s framing to better reflect its integration of foundational and contemporary research. We adjusted for relevant references, such as "Correale, J. Arias, M.; Gilmore, W. Steroid hormone regulation of cytokine secretion by proteolipid protein-specific CD4+ T cell clones isolated from  multiple sclerosis patients and normal control subjects. J Immunol 1998, 161 (7), 3365-3374." to "Hospital Espanol de Pachuca Research, G.; D. Licona-Menindez, D. Licona-Menindez, D. Licona-Menindez. Peon, A. N. Anti-COVID-19 Vaccination Alters the Menstrual Cycle and Dose Accumulation Enhances the Effect. Medicina (Kaunas)  2024, 60 (6). DOI: 10.3390/medicina60060956 From NLM."; "Penit, C.; Vasseur, F. Cell proliferation and differentiation in the fetal and early postnatal mouse thymus. J Immunol 1989, 142, 3369-3377" to. "Lavaert, M.; Valcke, B.; Vandekerckhove, B.; Leclercq, G.; Liang, K.L.; Taghon, T. Conventional and Computational Flow Cytometry Analyses Reveal Sustained Human Intrathymic T Cell Development From Birth Until Puberty. Front Immunol 2020, 11, 1659, doi:10.3389/fimmu.2020.01659."; "Utsuyama, M.; Hirokawa, K. Hypertrophy of the thymus and restoration of immune functions in mice and rats by gonadectomy. Mech Ageing Dev 1989, 47, 175-185, doi:10.1016/0047-6374(89)90030-4." to "Leposavić, G.; Obradović, S.; Kosec, D.; Pejcić-Karapetrović, B.; Vidić-Danković, B. In vivo modulation of the distribution of thymocyte subsets by female sex steroid hormones. Int Immunopharmacol 2001, 1, 1-12, doi:10.1016/s1567-5769(00)00006-0.".

Point 2: add the plan of the review at the end of Introduction; also data bases used, key words, inclusion criteria for the papers, time period for this research;

Response 2: Thank you for your advice. We have add the plan of the review at the end of Introduction: We systematically searched PubMed, Web of Science, and CNKI using keywords “thymus involution”, “gender differences”, and “hormonal regulation “ (1967–2025).

Inclusion criteria: (1) studies focusing on female thymus; (2) mechanistic insights into cellular/hormonal pathways; (3)  clinical or animal model data.

Point 3: indicate, pelase, for the Fig. 3, from which source this picture is adopted;

Response 3: Thank you for your careful review and for raising this important clarification.   We sincerely appreciate your diligence in ensuring the manuscript’s transparency and accuracy.

Figure 3 is an original schematic diagram created by our team to visually summarize the key mechanisms discussed in the manuscript. It was designed based on synthesized findings from multiple studies cited in the text.

Point 4: ! thymus is developed from 6 different types of cells. You cant explain the involution without analysis whats going on with each of these types in thymus. Thus, please, include a subsection with this description. Whats going on? In men, female, are there differences? Which cells suffers the most?  From which cells does the involution start? whats happening with this involution in ontogenetic aspect (during female life) etc? Thank you.

Response 4: Thank you for your advice. Because this paper mainly explores thymocytes, especially thymic epithelial cells, it is lack of comprehensiveness. the thymus is first a disproportionate loss of double positive (DP) cells and the differentiation of thymic epithelial progenitor cells (TEPC) accompanied by reduced turnover of TECs. there was an exclusive and concurrent increase in autoimmune regulator (Aire)+ cTEC/thymocyte ratio of autoimmune regulator (AIRE) middle-aged female mice with decrease in Aire+ medullary thymic epithelial cells (mTEC)/thymocyte ratio. Female  individuals exhibit enhanced local interactions,  specifically increased cell-cell interactions between cTECs and class II+ DP thymocytes,  as well as within the positive selection niche that includes dendritic cells (DCs) and DPs.  Menopausal thymus regeneration may expand this process, so exploration of this may bring new prospects for delaying thymic aging in women.

Round 2

Reviewer 1 Report

Comments and Suggestions for Authors

The Manuscript is improved.

Comments on the Quality of English Language

Minor editing of English Language required.

Reviewer 2 Report

Comments and Suggestions for Authors

Reviewer 3 Report

Comments and Suggestions for Authors

The author responded to my comments satisfactorily.

Reviewer 4 Report

Comments and Suggestions for Authors

Dear Authors,

Well, not all is improved, how I ment, but the existing improvement is also acceptable. And the manuscript commonly is an interesting issue. Thus, I will advice to publish it!